# Gas Chromatography Multiresidue Method for Enantiomeric Fraction Determination of Psychoactive Substances in Effluents and River Surface Waters

Ivan Langa [1,2], Maria Elizabeth Tiritan [1,2,3,4,*], Diana Silva [1,2,5] and Cláudia Ribeiro [1,2,3,*]

1 CESPU, Instituto de Investigação e Formação Avançada em Ciências e Tecnologias da Saúde, Rua Central de Gandra, 1317, 4585-116 Gandra, Portugal; langacry@gmail.com (I.L.); diana.silva@iucs.cespu.pt (D.S.)
2 TOXRUN—Toxicology Research Unit, University Institute of Health Sciences, CESPU, CRL, 4585-116 Gandra, Portugal
3 Centro Interdisciplinar de Investigação Marinha e Ambiental (CIIMAR/CIMAR), Universidade do Porto, Edifício do Terminal de Cruzeiros do Porto de Leixões, Av. General Norton de Matos s/n,o 4050-208 Matosinhos, Portugal
4 Laboratório de Química Orgânica e Farmacêutica, Departamento de Ciências Químicas, Faculdade de Farmácia, Universidade do Porto, Rua de Jorge Viterbo Ferreira, 228, 4050-313 Porto, Portugal
5 UCIBIO, REQUIMTE, Laboratory of Toxicology, Faculty of Pharmacy, University of Porto, R. Jorge Viterbo Ferreira, n° 228, 4050-313 Porto, Portugal
* Correspondence: elizabeth.tiritan@iucs.cespu.pt (M.E.T.); claudia.ribeiro@iucs.cespu.pt (C.R.)

**Abstract:** Determination of psychoactive substances (PAS) and/or their metabolites in surface waters is crucial for environmental risk assessment, and disclosure of their enantiomeric fractions (EF) allows discrimination between consumption, direct disposal, and synthesis pathways. The aim of this study was to develop and validate an indirect method by gas chromatography coupled to mass spectrometry (GC–MS) based on derivatization using (R)-(−)-α-methoxy-α-(trifluoromethyl) phenylacetyl chloride as chiral derivatization reagent, for enantiomeric quantification of amphetamine (AMP), methamphetamine (MAMP), 3,4-methylenedioxymethamphetamine (MDMA), norketamine, buphedrone (BPD), butylone, 3,4-dimethylmethcathinone (3,4-DMMC), 3-methylmethcathinone, and quantification of 1-benzylpiperazine and 1-(4-metoxyphenyl)-piperazine. The method allowed to evaluate the occurrence, spatial distribution, and the EF of the target chiral PAS in Portuguese surface waters and in effluents from 2 wastewater treatment plants (WWTP). For that, water samples were pre-concentrated by solid phase extraction using OASIS® MCX cartridges, derivatized and further analyzed by GC–MS. Both enantiomers of AMP, (R)-MDMA, (S)-MAMP, and the first eluted enantiomer of BPD (configuration not assigned) were found in surface waters, while effluent samples showed both enantiomers of MDMA, (S)-MAMP, (R)-AMP, and the first eluted enantiomer of BPD and 3,4-DMMC. According to our knowledge, this is the first multiresidue analytical method by CG–MS enrolling cathinones, amphetamines, and piperazines. The presence of illicit synthetic cathinones in Douro River estuary is here reported for the first time, along with other amphetamine derivatives. The potential of the method to monitor consumption of the target PAS was demonstrated.

**Keywords:** synthetic cathinones; amphetamines; estuarine water; psychoactive drugs; enantioselective; wastewater treatment plants

## 1. Introduction

The misuse of psychoactive substances (PAS) has been reported all over the world with consequent negative social, economic, and public health problems. According to the European Monitoring Centre for Drugs and Drug Addiction (EMCDDA), new psychoactive substances (NPS), such as synthetic amphetamines (AMPs) and cathinones (SCAT), the second major group of NPS, were reported in a rate of more than 50 new drugs per year since 2012, reaching a peak in 2014–2015 (close to 100 NPS). In 2020, only 46 NPS

were first time reported in Europe representing a reduction of approximately 50% of the previous peak [1]. The possible reason for this reduction is the introduction of the new general prohibitions and regulation/legislation on generic and analogous substances, which occurred throughout the Europe, including Portugal [2–4]. Nevertheless, the number of the NPS remains worrisome [2,3]. Most of these new drugs are produced by chemical and pharmaceutical companies in Asian countries and sold worldwide [3], although Europe is also a production region, favoring their market and access by the local consumers.

Measuring drug consumption patterns is difficult but essential for health-care professionals, risk assessment authorities and policy-makers.

Many PAS are chiral and, depending on the manufacturing procedure, may be available either as racemate or as single enantiomers. On the other hand, human metabolism or transformation processes can further lead to a racemization or enantiomeric enrichment [5–7], turning the determination of the enantiomeric fraction (EF) essential for as accurate analysis of these drugs [5,8,9].

The biodegradation of PAS and their metabolites during wastewater treatment can also affect the EF [10–12]. Therefore, PAS and their metabolites are frequently detected in effluents and surface waters, at different EF of the excretion. As such, enantioselective analysis of PAS in WWTP effluents and surface waters is of high importance for risk assessment as enantiomers may display different impact on non-target organisms [9,13].

Regarding enantioselective methods for PAS analysis, liquid chromatography (LC) and gas chromatography (GC) are among the most used [9,14] for water matrices such as wastewaters and surface waters [7,15,16]. The enantioseparation can be done by a direct method using chiral stationary phases [9,17] or an indirect method through formation of diastereomers by reaction with an enantiomerically pure reagent [7,16].

The main purpose of this study was to develop a method by GC–MS based on the formation of diastereomers for quantification of several classes of PAS such as AMPs [amphetamine (AMP), methamphetamine (MAMP), 3,4-methylenedioxymethamphetamine (MDMA)], SCAT [buphedrone (BPD), butylone (BTL), 3,4-dimethylmethcathinone (3,4-DMMC) and 3-methylmethcathinone (3-MMC)], the ketamine metabolite, norketamine (NK), and derivatives of piperazine (PP) 1-BP and 1,4-MPP, using the enantiomerically pure reagent $(R)$-$(-)$-$\alpha$-methoxy-$\alpha$-(trifluoromethyl) phenylacetyl chloride [$(R)$-MTPA-Cl]. These substances were selected based on EMCDDA reports and literature regarding the consumption of NPS [1,2,18]. To the best of our knowledge, there are not reports for the enantioselective determination of all these classes of compounds in surface waters. The method was applied to investigate the occurrence and spatial distribution of the selected PAS in Portuguese surface waters of Douro River estuary and effluent samples from two WWTPs with different treatment technologies.

## 2. Materials and Methods

### 2.1. Chemicals and Materials

AMP and MAMP standards were acquired from Lipomed (Arlesheim, Switzerland); 1-BP from Chemos GmbH (Regenstauf, Germany); 1,4-MPP from Acros Organics (Morris Plains, NJ, USA); 3-MMC and 3,4-DMMC from LGC Standards GmbH (Wesel, Germany); and BPD from Cayman Chemical (Ann Arbor, MI). NK was purchased from Sigma Aldrich (Steinhein, Germany); and BTL and MDMA from Cerilliant (Round Rock, TX, USA). $(d,l)$-AMP-d$_3$ was purchased from Lipomed (Arlesheim, Switzerland) and used as internal standard (IS) for AMPs and SCATs. The chiral reagent $(R)$-MTPA-Cl, triethylamine (TEA), ammonium hydroxide 25% (NH$_4$OH), sodium hydroxide (NaOH), and sulfuric acid (H$_2$SO$_4$) were purchased from Sigma-Aldrich (Steinheim, Germany). Individual stock solutions of standards were prepared in methanol (MeOH) at 1 mg mL$^{-1}$ and stored at $-20\ ^{\circ}$C in amber vials. All reference standards were >98% pure. Log K$_{ow}$ and p$K$a values can be found in Table S1. Work solutions were prepared freshly by dilution of stock solutions in MeOH. Stock solutions of IS were prepared at 200 µg mL$^{-1}$. The chiral reagent solution

was obtained by dilution of 5 µL of (*R*)- MTPA-Cl in 95 µL of anhydrous acetonitrile (ACN) and stored in amber vials at −20 °C.

All solvents used were of HPLC grade. ACN, *n*-hexane (Hex), MeOH, and ethanol (EtOH) were purchased from VWR Prolab Chemical (Radnor, PA, USA). Anhydrous ACN and anhydrous ethyl acetate were purchased from Merck (Darmstadt, Germany). Formic acid 98–100% was purchased from Merck (Espoo, Finland). Ultra-pure water was supplied by a SG Water System (Ultra Clear UV model). Glass microfibers filter with 0.7 µm pore size was purchased from VWR (Leuven, Belgium). Two-milliliter syringes were purchased from BD Emerald (Madrid, Spain). Syringe filters with 0.22 µm pore size were purchased from Teknokroma (Barcelona, Spain). Oasis MCX 150 mg (6 cc) solid phase extraction (SPE) cartridges were purchased from Waters (Dublin, Ireland).

### 2.2. Equipment

#### 2.2.1. Chromatographic System

Chromatographic analysis was performed using a Varian CP-3800 gas chromatography equipped with ion-trap Varian Saturn 2200 mass detector and electron impact (EI) ionization chamber, an autosampler (Varian CP-8400), and an electronically controlled split/splitless injection port. Chromatographic separation was achieved using a Zebron (5% phenyl, 95% dimethylpolysiloxane) capillary column (30 m × 0.25 mm I.D. × 0.25 µm film thickness), from Phenomenex, USA. High-purity helium (99.999%) was used as carrier gas.

#### 2.2.2. Other Equipment

A centrifugal vacuum evaporator (CentriVap Concentrator) with a cold trap purchased from Labconco (Kansas City, MO, USA) was used to evaporate sample extracts. Visiprep™ SPE Vacuum Manifold purchased from Supelco was used for SPE procedure. A multiparameter analyzer Consort C863 (Turnhout, Belgium) was used for determination of physico-chemical parameters of estuarine water samples: pH, total dissolved solids (TDS), and electrical conductivity (EC).

### 2.3. Sample Collection

For method validation, spring water samples from the source of the Leça River were collected and used as blank matrix (to compensate matrix effects and allow an accurate quantification of the analytes). Upon arrival to the laboratory, all samples were immediately vacuum filtered through a 0.7 µm glass fiber filter, acidified to pH ≈ 3 with $H_2SO_4$ (95–97%), and stored into amber glass bottles at 4 °C in the dark.

For method application, 1 L of estuarine water was collected at 5 sampling points (S1–S5) of Douro River estuary, from the river outlet, near the Atlantic Ocean, to the Crestuma-Lever dam (Figure 1), in summer (August 2020). Sampling stations S1 (mouth of the river Douro), S3 (Freixo) and S4 (mouth of the Sousa River) are located on the north bank of the river at the Porto city margin, whereas S2 (mouth of the river Douro) and S5 (Crestuma-Lever dam) are located at the opposite side, bordering the other highly industrialized and densely inhabited region, the Vila Nova de Gaia city. Temperature was measured in situ at each sampling point. During the transport, samples were kept refrigerated (±4 °C) in the dark. Upon arrival at the laboratory, pH, EC, and TDS were immediately measured, and samples were vacuum filtered through 0.7 µm glass fiber filters to remove suspended particles, acidified to pH 3 with $H_2SO_4$ (conc.), and stored at ±4 °C in the dark, for a maximum period of 12 h until SPE procedure.

Furthermore, 24 h-composite samples of 2 WWTPs (WWTPA and WWTPB) located in the Greater Porto region with different treatment processes were collected using amber bottles, in summer (9 July) of 2020. WWTPA receives urban wastewater and wastewater treatment consists of a biological treatment with conventional activated sludge system operating under aeration regime. WWTPB mainly receives urban wastewater and performs both secondary biological treatment with activated sludge system and tertiary treatment by UV light. Both WWTPs effluents are discharged into tributaries of Douro River.

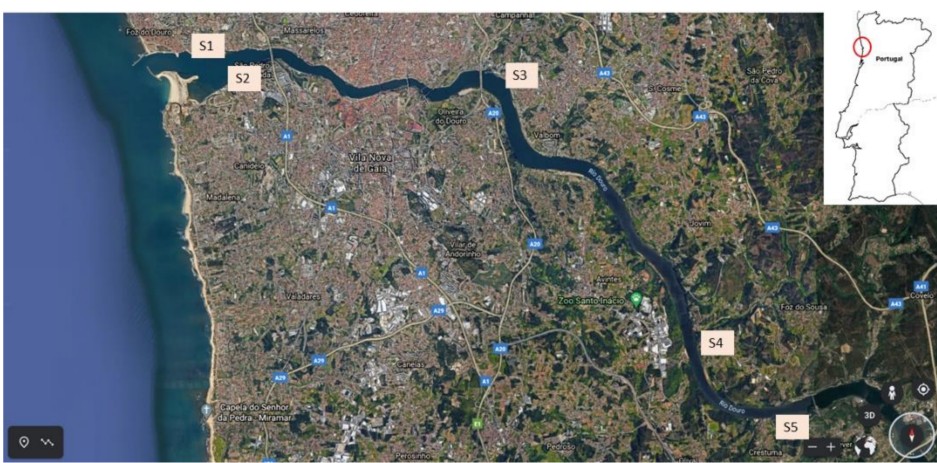

**Figure 1.** Location of the five sampling points (S1 to S5) of Douro River estuary, Portugal.

### 2.4. Solid-Phase Extraction (SPE) Procedure

The SPE procedure was adapted from that described by Coelho et al. [15]. Briefly, for method optimization and validation, 1 L of pre-filtered and acidified spring water was spiked with 250 μL of mixture of the standards of the target compounds at different ranges of concentrations (Tables S2 and S3) and the IS at 500 ng mL$^{-1}$. For method application, 1 L of estuarine water samples or effluent samples was spiked with the IS at 500 ng mL$^{-1}$.

The SPE was performed using a Visiprep$^{TM}$ SPE Vacuum Manifold and OASIS$^{®}$ MCX cartridges (150 mg, 6 cc), without cartridge conditioning. Samples were directly loaded into the cartridges at a flow rate of 5 mL min$^{-1}$. Then, cartridges were washed with 4 mL of 2% formic acid. After washing, cartridges were dried under vacuum for 1 h. The elution was performed with 4 mL of 5% NH$_4$OH in EtOH. The eluates were then filtered, with a 0.22 μm syringe filter previously rinsed with 1 mL of 5% NH$_4$OH in EtOH. After filtration, the syringe filters were washed with 1 mL of EtOH to ensure maximum recovery of the analytes. Eluates were evaporated to dryness using a centrifugal vacuum evaporator and then reconstituted in 250 μL of MeOH. After the SPE, the extracts were derivatized according to the following derivatization procedure.

### 2.5. Derivatization with Chiral Derivatization Reagent

The derivatization procedure described by Gonçalves et al. (2019) was adapted and used for the formation of the diastereomers of 8 chiral PAS (4 SCAT, 3 AMP-like substances and NK), using (*R*)-MTPA-Cl as chiral derivatization reagent [7]. Piperazines are not chiral, however, 2 PP derivatives were also obtained.

For the derivatization of the target compounds, 200 μL of standard mixtures or aliquots of SPE extracts were transferred into a vial and evaporated to dryness in a speedvac, at room temperature. Then, 200 μL of ultra-pure water and 200 μL of NaOH (1 M) were added to the residue and vortexed for 30 s. After that, 1500 μL of 0.02% TEA in Hex were added, the solution was vortexed for 10 min for compounds extraction, and then centrifuged at 13,000 rpm for 10 min for the phase separation. Then, 1200 μL of the organic phase was transferred to a new vial and 10 μL of chiral reagent solution were added, followed by 2 h of heating at 80 °C. After that, the samples were cooled to room temperature and 100 μL of EtOH was added. The solution was heated in the oven at 70 °C for 15 min to stop the derivatization reaction. Finally, the samples were cooled and evaporated to dryness in the speedvac. The residue was reconstituted in 200 μL of anhydrous ACN and analyzed by GC–MS (1 μL).

### 2.6. Chromatographic Conditions of the GC-MS

Different conditions were attempted to optimize separation of the diastereomers and PP derivatives, using standard mixtures. Optimized chromatographic conditions were

achieved using a Zebron capillary column at a constant flow rate of 1.0 mL min$^{-1}$. The injection port temperature was programmed at 280 °C and 1 μL of sample was injected in splitless mode. The oven temperatures were programmed as follows: an initial temperature of 140 °C was held for 50 s, followed by a ramp to 215 °C at 11 °C min$^{-1}$ and held for 5 min, a ramp to 285 °C at 10°C min$^{-1}$, and maintained for 20 min, with a total run time of 24.32 min. The MS operating conditions were EI mode with electron energy of 70 eV, operated in both full scan (FS) mode from $m/z$ 40 to 650 (total ion count, TIC) and selection ion storage (SIS) according to their $m/z$ fragments obtained from the MS of each target derivatives. To determinate the order of elution of the diastereomers, individual standards at 10 μg mL$^{-1}$ of (S)-AMP, (R)-MAMP, (S)-MDMA were derivatized according to the procedure described in Section 2.5 [7].

### 2.7. Method Parameters and Validation

The method was validated according to the International Conference on Harmonization (ICH), considering the following parameters: selectivity, linearity, limit of detection (LOD), limit of quantification (LOQ), accuracy, precision, and recovery [19].

Selectivity was verified by comparing the chromatograms in both FS and SIS modes of MS spectra of the solvent standard mixtures, spiked and non-spiked extracted samples from spring water samples (used as blank matrix) from the source of Leça River.

Linearity was studied using matrix-matched calibration by spiking 1 L of blank matrices, i.e., spring waters from source of Leça River, at five or six calibration standards mixtures, each one in triplicate, containing the IS. The range of concentrations of each PAS are present in Table S2. The calibration curve linearity was evaluated by its correlation coefficient (r$^2$). For AMP-like substances and SCAT, equations were obtained after least-squares linear regression of the ratio analyte/IS. For NK and PP derivatives, no IS was used, and calibration curves were obtained by external standard calibration.

The LOD and LOQ were determined based on the standard deviation of the response and the slop. The following equations were used to calculate LOD and LOQ [19,20]:

$$\text{LOQ} = 10 \times (s/S), \text{LOD} = 3.3 \times (s/S)$$

where $s$ = Standard deviation of the response and $S$ = Slop of the calibration curve.

For accuracy, intra- and inter-precision and recovery assays, three quality controls (QCs) standard solutions covering the dynamic linear range (low, medium, and high) were added to blank water samples, each one prepared and analyzed in triplicate and this procedure was performed in three different days. The selected QC concentrations for each PAS are present in Table S3.

Accuracy was determined as the percentage of agreement between the method results and the nominal amount of added compound, using the following equation:

$$\text{Accuracy } (\%) = (\text{Real Conc.}/\text{Nominal Conc.}) \times 100\%$$

Precision was expressed by the relative standard deviation (% RSD) of the replicate measurements.

Recovery was calculated by the ratio of peak area of analyte/peak area of IS or peak area of analyte for NK and PP obtained after SPE procedure of blank water samples previous spiked with a standard mixture and a solvent standard mixture peak area of analyte/peak area of IS or peak area of analyte for NK and PP at the same concentrations. EF was used to express the relative concentration of diastereomers. When the configurations of the eluted enantiomers were known, as for the cases of AMP, MAMP, and MDMA according to previous report [7], they were assigned as (S) and (R) for each enantiomer. [D1] and [D2] were used to designate the concentrations of the first and second diastereomers eluted from the column, respectively, when elution order was unknown. The following equations were used for calculation of EF [21]:

$$\text{EF} = \frac{[S]}{([S] + [R])}$$

$$EF = \frac{[D1]}{([D1] + [D2])}$$

## 3. Results and Discussion

### 3.1. Derivatization with Chiral Derivatization Reagent

The enantiopure derivatization reagent (*R*)-MTPA-Cl was used for the formation of the diastereomers of AMP-type substances, SCAT and NK. In a previous study done by our research group, (*R*)-MTPA-Cl was selected as the best derivatizing reagent for the formation of diastereomer of PAS, such as AMP and NK [7]. Nevertheless, this procedure was never reported for the formation of diastereomers of SCAT. Therefore, the derivatization procedure was optimized for the reaction with substances of these class, allowing the formation of the diastereomers that were confirmed by the respective MS. The products of reaction with (*R*)-MTPA-Cl of all target compounds are presented in Table S4.

The enantiomers of AMP-type substances SCAT and NK are converted into diastereomers by formation of amides by *N*-acylation (Figure 2). Furthermore, it was possible to observe the formation of PP derivatives, confirmed by the respective MS. The formation of these PP derivatives enhanced not only the sensitivity of the method, but also allowed the use of other *m/z* fragments to confirm the occurrence of these PP in complex matrices such as environmental matrices.

Figure 2 displays the scheme of reaction of (*R*)-MTPA-Cl with the enantiomers of AMP (Figure 2a), BPD (Figure 2b), and NK (Figure 2c) to achieve the diastereomers as well as the reaction with 1-BP (Figure 2d).

(*R*)-MTPA-Cl

AMP

(*R*)-3,3,3-trifluoro-2-methoxy-2-phenyl-*N*-((*S*)-1-phenylpropan-2-yl)propanamide

(*R*)-3,3,3-trifluoro-2-methoxy-2-phenyl-*N*-((*R*)-1-phenylpropan-2-yl)propanamide

(**a**)

**Figure 2.** *Cont.*

(R)-3,3,3-trifluoro-2-methoxy-N-methyl-N-((S)-1-oxo-1-phenylbutan-2-yl)-2-phenylpropanamide

(R)-3,3,3-trifluoro-2-methoxy-N-methyl-N-((R)-1-oxo-1-phenylbutan-2-yl)-2-phenylpropanamide

(**b**)

(R)-N-((S)-1-(2-chlorophenyl)-2-oxocyclohexyl)-3,3,3-trifluoro-2-methoxy-2-phenylpropanamide

(R)-N-((R)-1-(2-chlorophenyl)-2-oxocyclohexyl)-3,3,3-trifluoro-2-methoxy-2-phenylpropanamide

(**c**)

(R)-1-(4-benzylpiperazin-1-yl)-3,3,3-trifluoro-2-methoxy-2-phenylpropan-1-one

(**d**)

**Figure 2.** Reaction of (R)-MTPA-Cl with the enantiomers of AMP (**a**), BPD (**b**), NK (**c**) for diastereomer formation and 1-BP, and (**d**) for the parent drug derivative formation. * refers to the stereogenic center.

*3.2. Optimization of the Chromatographic Separation of the Diastereomers and Piperazine Derivatives*

Different conditions were attempted to optimize separation of the diastereomers and PP derivatives, using standard mixtures. For that, different ramps of temperature, EI ionization voltages and flow rate of the carrier gas were tested. After various attempts,

the optimized conditions (already described in Section 2.6) allowed the separation of the diastereomers of all AMP-type substances (AMP, MAMP, MDMA), NK and PP derivatives (1-BP and 1,4-MPP). Considering SCAT, separation of diastereomers were achieved for BTL and 3,4-DMMC. Furthermore, diastereomer resolution was possible for both BPD and 3-MMC; nevertheless, co-elution of the second BPD diastereomer (D2) and first 3-MMC diastereomer (D1) was observed in the standard mixture. Various attempts were made for a better resolution but without success. However, identification and quantification of both compounds were possible due to some differences in the respective MS.

Figure 3 shows the chromatograms of a standard mixture at 1 μg mL$^{-1}$ of all the target PAS with the separation of AMP, MAMP, MDMA, NK, BPD, 3-MMC, 3,4-DMMC, and BTL diastereomers, as well as the derivatized non-chiral PP (1-BP and 1,4-MPP).

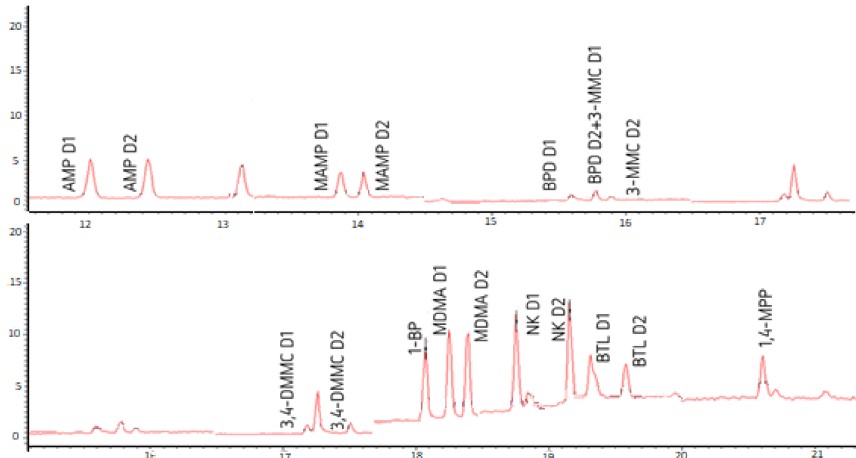

**Figure 3.** Chromatograms of a standard mixture at 1 μg mL$^{-1}$, showing the separation of the diastereomers of the amphetamines: amphetamine (AMP), methamphetamine (MAMP), and 3,4-methylenedioxy-*N*-methylamphetamine (MDMA); norketamine (NK); synthetic cathinones: buphedrone (BPD), 3-methylmethcathinone (3-MMC), 3,4-dimethylmethcathinone (3,4-DMMC) and butylone (BTL); as well as piperazines derivatives: 1-benzylpiperazine (1-BP) and 1-(4-metoxyphenyl) piperazine (1,4-MPP). D1 and D2 correspond to the first and second eluted diastereomers, respectively.

*3.3. Mass Spectra of the Target Compound Diastereomers and Piperazine Derivatives*

For identification of the compounds, individual standards at a concentration of 1 μg mL$^{-1}$ were derivatized, according to the procedure already described in Section 2.5.

Acquisition of the MS of each diastereomer and PP derivative was performed in FS mode. Table 1 shows the characteristic fragmentation ions (*m/z*) for identification, the quantification ions (QI), as well as the retention time (RT) of each diastereomer. Figures 4–7 show the chromatograms and MS with the possible fragmentation pattern of each target PAS.

For the IS AMPd$_3$, the fragments *m/z* 235 and 263 were the most abundant and were used for both identification and quantification.

The fragment 189 (*m/z*) was observed in the MS of all diastereomers (Table 1 and Figures 4–7). This fragment is characteristic of the reagent (*R*)-MTPA-Cl [7,22,23] and therefore was not used for quantification.

The molecular ions [M]$^+$ of AMP, MAMP, and MDMA were barely detectable and therefore of little qualitative and quantitative value.

It was possible to observe in the MS spectra from AMP and MAMP the presence of the fragments 91 and 119 *m/z*. Fragment at *m/z* 91 could be a result of benzyl cation ([C$_7$H$_7$]$^+$). The mass spectra showed high relative abundance ions at *m/z* 119 corresponding to the phenylpropane hydrocarbon radical cation that is the pharmacophore of both AMP-type substances and SCAT.

NK presented a different fragmentation pattern from the amphetamines due to the differences in their scaffold. Among abundant fragments were *m/z* 207 and 250 (Figure 6 and Table 1).

Although the PP are not chiral, the presence of the ion 189 *m/z* was observed in both 1-BP (Figure 7, Table 1) and 1,4-MPP (Table 1) showing the reaction of piperazines with the derivatization reagent (*R*)-MTPA-Cl. Although there is no diastereomer formation, it allowed to improve the signal identification and detection of both compounds. MS spectra of piperazine derivatives also showed the abundance of the fragment 91 *m/z* corresponding to the loss of the benzyl group (Table 1 and Figure 7).

The chemical structure of both PP (1-BP and 1,4-MPP) are quite similar, differing on the methoxy group in *para* position of the benzyl group (1,4-MPP). This similarity corroborates the presence of the ion 392 *m/z* for 1-BP corresponding to the molecular ion [M⁺] of this derivative, and the fragment at *m/z* 393 corresponding to the loss of the methoxyl (-OCH₃) in the 1,4-MPP derivative.

**Table 1.** Characteristic fragmentation ions (*m/z*), quantification ions (*m/z*) and retention time of the diastereomers of the target compounds.

| Compound | *m*/z | QI | RT (Minutes) | |
|---|---|---|---|---|
| | | | D1 | D2 |
| AMPd3 | 92; 119; 165; 189; 235; 263 | 235; 263 | 12.02 | 12.45 |
| AMP | 91; 119; 162; 189; 234; 260 | 162; 234; 260 | 12.04 * | 12.47 ** |
| MAMP | 91; 119; 148; 176; 189; 274 | 274 | 13.88 ** | 14.05 * |
| BPD | 105; 119; 189; 288 | 288 | 15.47 | 15.60 |
| 3-MMC | 119; 189; 274 | 274 | 15.60 | 15.89 |
| 3,4-DMMC | 105; 119; 133; 189; 200; 274 | 274 | 17.18 | 17.50 |
| 1-BP | 91; 175; 189; 392 | 392 | 18.06 | |
| MDMA | 119; 135; 162; 189; 274 | 162; 274 | 18.25 * | 18.39 ** |
| NK | 189; 207; 250; 404 | 206; 250 | 18.75 | 19.16 |
| BTL | 119; 149; 189; 207; 288 | 288 | 19.32 | 19.58 |
| 1,4-MPP | 91; 189; 207; 393; 408 | 408 | 20.61 | |

*: (*R*) enantiomer; **: (*S*) enantiomer.

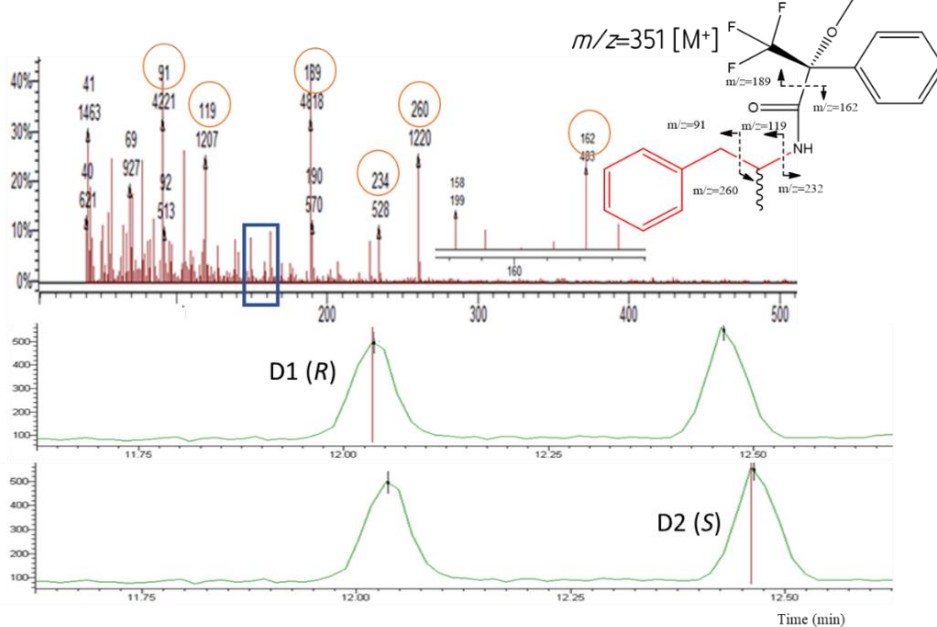

**Figure 4.** Chromatograms showing the AMP diastereomers D1 [(*R*)-AMP] and D2 [(*S*)-AMP], respectively (**bottom**); the MS and the possible fragmentation pattern (**top**).

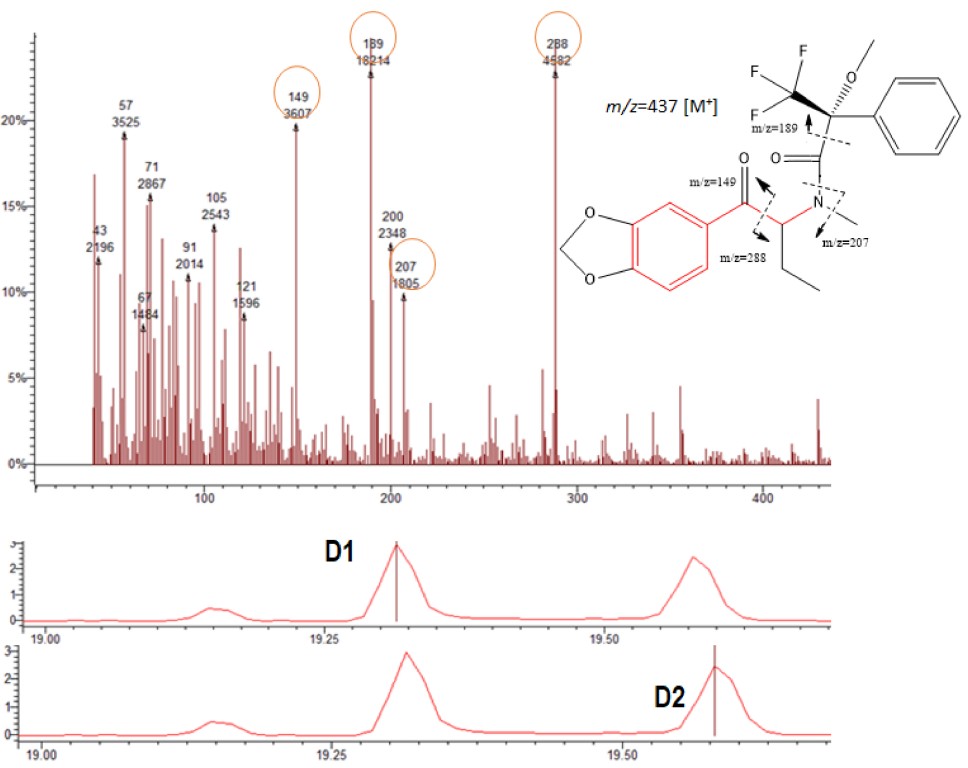

**Figure 5.** Chromatograms showing the BTL diastereomers D1 and D2, respectively (**bottom**); mass spectrum and the possible fragmentation pattern (**top**).

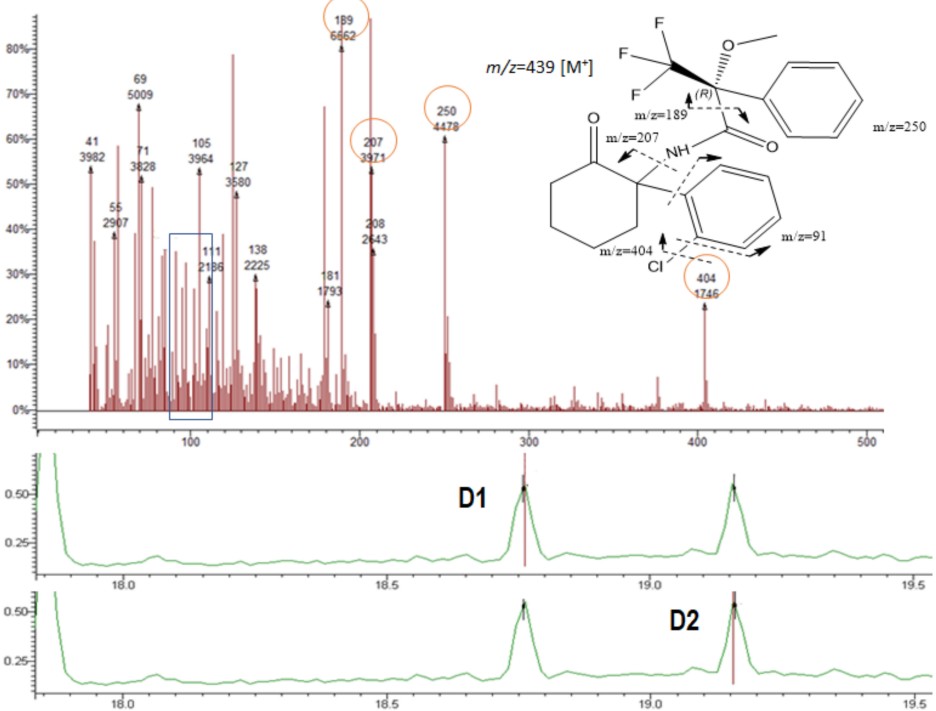

**Figure 6.** Chromatograms showing the NK diastereomers D1 and D2, respectively (**bottom**); mass spectrum and the possible fragmentation pattern (**top**).

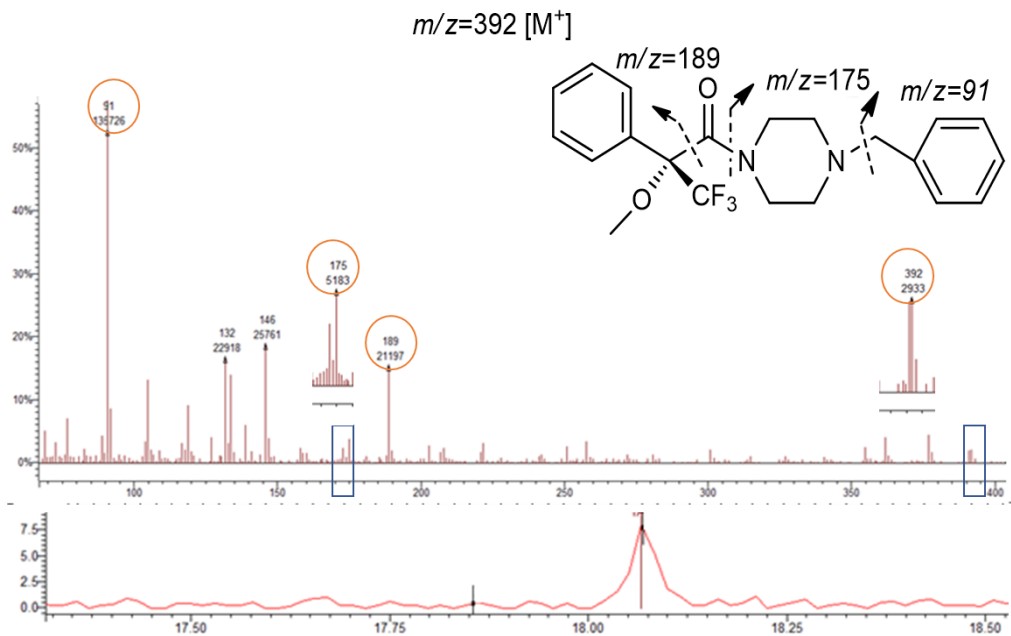

**Figure 7.** Chromatogram of 1-BP derivative (**bottom**); mass spectrum and suggested fragmentation pattern (**top**).

### 3.4. Method Validation

Validation of the analytical method was performed according to ICH guidelines and considering the following parameters: selectivity, linearity, accuracy, precision, recovery, LOQ, and LOD. Due to the coelution of AMP/AMP $d_3$ (IS), BPD D2/3-MMC D1 and to a matrix effect on 3,4-DMMC D2, no major gain in SIS mode, selectivity, LOD, and LOQ was verified comparatively to FS mode, for these compounds.

Therefore, MS detection was conducted in both FS mode, from which the quantification ions were selected and extracted (according to software program), or SIS depending on the optimized conditions for each target PAS. Selectivity was verified by comparing the chromatograms of solvent standard mixture, spiked and non-spiked extracted samples from spring water samples (used as blank matrix) from the source of Leça River. It was verified that the analytical method was selective for the quantification of all the target compounds (Figure 8).

For linearity, a range of 6 concentration levels for AMP, MDMA, and BTL, and 5 concentration levels for MAMP, BPD, 3-MMC, 3,4-DMMC, NK, 1,4-MPP, and 1-BP were performed considering the LOQ as the first point of each calibration curve (Table 2).

The method showed to be linear with the $r^2$ ranging from 0.9846 to 0.9972 for all target compounds; the method LOD was from 14.2 to 89.5 ng $L^{-1}$ and the method LOQ between 50.0 and 250 ng $L^{-1}$ (Table 2).

Regarding accuracy determination, the ICH guidelines recommend ranges to be considered from 80 to 120% of the test concentration. In this study, accuracy values ranged from 82.4 to 116.9%, showing that the method presents accuracy within acceptable values established by ICH (Table 2).

The precision was estimated by calculating the relative standard deviation (% RSD). Values were lower than 7.83%, which are in accordance with those demanded by ICH (under 15%). Recoveries were reproducible and varied between 18.6% and 98% (3-MMC and (*R*)-MAMP), respectively. The dissimilar recoveries are owing to the wide range of target compounds and were taken into account using matrix match calibration curves.

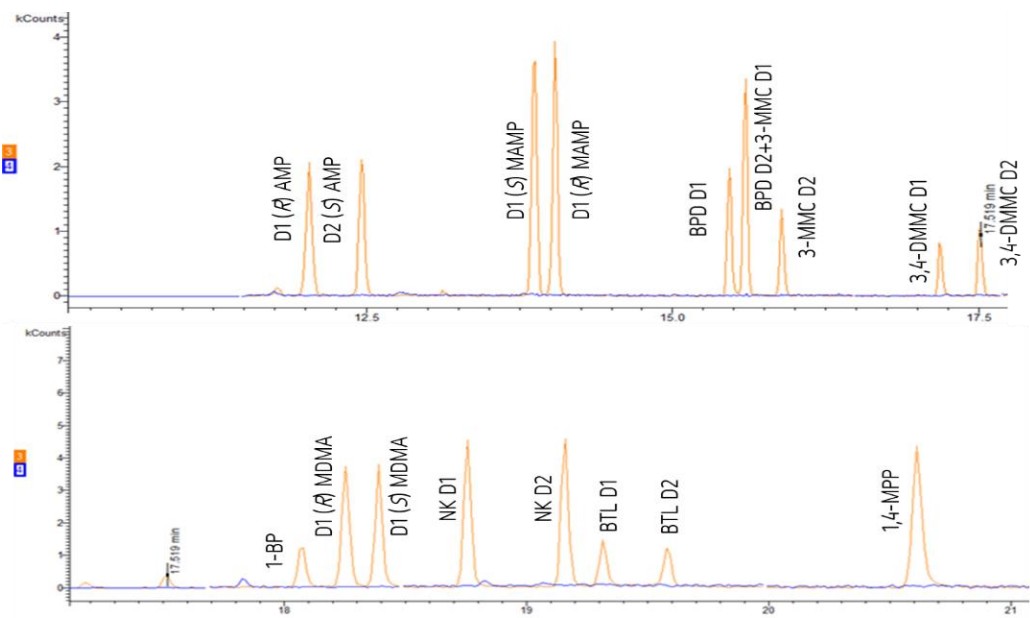

**Figure 8.** Chromatogram with comparison of the blank (blue) and the standard mixture (orange) at 1 μg mL$^{-1}$.

**Table 2.** Linearity parameters, method LOD and LOQ, recovery (%), accuracy (%), and precision (% RSD).

| PAS | Concentration Range (ng L$^{-1}$) | Equation | r$^2$ | LOD (ng L$^{-1}$) | LOQ (ng L$^{-1}$) | Recovery (%) | Accuracy (%) | Precision RSD (%) |
|---|---|---|---|---|---|---|---|---|
| AMP (*R*) | 50.0–300 | y = 0.0071 (±0.00037)x + 0.5269 (±0.068) | 0.9891 | 31.8 | 50.0 | 84.5 | 89.4–107.0 | 2.39–5.31 |
| AMP (*S*) | 50.0–300 | y = 0.0071(±0.00065)x + 0.563 (±0.12) | 0.9846 | 38.0 | 50.0 | 83.5 | 88.4–108.7 | 3.65–5.53 |
| MAMP (*S*) | 50.0– 300 | y = 0.0081(±0.00026)x − 0.0145 (±0.044) | 0.9968 | 18.0 | 50.0 | 83.7 | 107.7–109.9 | 0.49–3.54 |
| MAMP (*R*) | 50.0–300 | y = 0.0081(±0.00037) + 0.1966 (±0.063) | 0.9935 | 25.0 | 50.0 | 98.0 | 96.6–106.6 | 0.87–3.42 |
| BPD D1 | 125–425 | y = 0.0017 (±0.00088)x + 0.0773 (±0.027) | 0.9938 | 35.9 | 125 | 48.5 | 85.8–104.0 | 3.07–5.60 |
| BPD D2 | 125–425 | y = 0.0018 (±0.00087)x + 0.0609 (±0.019) | 0.992 | 40.8 | 125 | 43.5 | 90.1–103.2 | 0.40–3.73 |
| 3-MMC D1 | 250–575 | y = 0.001 (±0.0009)x − 0.1153 (±0.037) | 0.9887 | 89.5 | 250 | 19.7 | 82.9–94.9 | 2.27–4.86 |
| 3-MMC D2 | 250–575 | y = 0.001 (±0.0009)x − 0.0899 (±0.029) | 0.9928 | 71.1 | 250 | 18.6 | 82.4–94.0 | 1.00–6.94 |
| 3,4-DMMC D1 | 250–625 | y= 0.0018 (±0.0001)x − 0.2648 (±0.044) | 0.9928 | 70.3 | 250 | 55.7 | 89.5–90.9 | 1.02–4.41 |
| 3,4-DMMC D2 | 250–625 | y= 0.0018 (±0.0001)x − 0.2585 (±0.038) | 0.9919 | 74.3 | 250 | 50.2 | 83.2–91.6 | 3.88–7.83 |
| MDMA (*R*) | 75.0–375 | y = 0.0108 (±0.00051)x − 0.23 (±0.122) | 0.9909 | 52.0 | 75.0 | 86.4 | 111.0–115.2 | 1.97–4.92 |
| MDMA (*S*) | 75.0–375 | y = 0.0108 (±0.00048)x − 0.1765 (±0.089) | 0.9949 | 38.0 | 75.0 | 88.7 | 109.9–116.9 | 2.14–4.92 |
| NK D1 | 75.0–375 | y = 53.535 (± 1.98)x − 1227.7 (±256) | 0.9972 | 14.2 | 75.0 | 76.1 | 87.3–108.4 | 3.90–5.64 |
| NK D2 | 75.0–375 | y = 56.272 (±0.98)x − 1679.8 (±176) | 0.9903 | 26.4 | 75.0 | 74.2 | 87.7–107.9 | 4.04–7.37 |
| BTL D1 | 75.0–375 | y = 0.0037 (±0.00037)x + 0.325 (±0.036) | 0.9926 | 24.7 | 75.0 | 78.1 | 88.9–97.9 | 0.98–4.84 |
| BTL D2 | 75.0–375 | y = 0.0038 (±0.00049)x + 0.2954 (±0.023) | 0.9906 | 17.0 | 75.0 | 82.0 | 88.2–101.8 | 1.96–5.73 |
| 1-BP | 250–625 | y = 70.682 (±4.46)x − 7237.6 (±1892) | 0.9882 | 88.0 | 250 | 53.0 | 97.3–112.4 | 3.27–7.35 |
| 1,4-MPP | 75.0–250 | y = 44.495 (±2.18)x − 2729.9 (±353) | 0.9928 | 29.0 | 75.0 | - | 98.9–112.9 | 2.69–5.14 |

*3.5. Application of the Method*

The method was applied to ascertain the occurrence, spatial distribution, and the EF evaluation of the target analytes in real surface waters and effluents from WWTPs collected in the area of the second largest Portuguese city, Porto. Five sampling points along the estuary of the Douro river were selected according to previous studies [15,24]. Douro river is the third-longest river in the Iberian Peninsula, and it has a watershed shared between Spain (80%) and Portugal (20%) [25], receiving directly or indirectly effluents of 8 WWTPs.

Physico-chemical parameters of the water samples collected were measured, and data are shown in Table S5. These were within values found in previous monitoring studies and expected values for estuarine water samples [24].

Regarding AMPs, both enantiomers of AMP were found though at <LOQ in S1, while only enantiomer (*S*)-MAMP was found at S1 and S4 and (*R*)–MAMP bellow LOD (Table S6). Sampling point S1 is located in the river mouth near the discharge of one of the highest WWTPs of Porto city, Sobreira WWTP. Sampling point S4 is also a hot spot, located at the mouth of Sousa River, a tributary of Douro River (Figure 1) and near the discharge of Sousa WWTP. Concerning AMP, distinction between consumption and direct disposal poses a

significant challenge due to legal and illicit use. In Portugal, only (*S*)-AMP is available as a prescription medication for the treatment of hyperactivity disorder and/or attention deficit. Therefore, it would be expected an enrichment of (*S*)-AMP, nevertheless, both enantiomers of AMP were at EF ~0.5 suggesting other sources of AMP.

Leuckart reaction is the most common synthetic route for illicit synthesis of AMP producing a racemate. After consumption, (*S*)-AMP is metabolized faster than (*R*)-AMP. Consequently, excreted AMP is enriched with (*R*)-AMP. Microbial processes during WWTP also favors (*S*)-AMP, and thus (*R*)-AMP is more recalcitrant.

Regarding MAMP, it is an illicit drug exclusively used in European countries, only (*S*)-MAMP (EF ≅ 1) was found, which is in accordance with our previous study [7]. The presence of (*S*)-MAMP shows illegal discharge or illicit consumption. In fact, most common synthesis process in central Europe uses *L*-ephedrine as starting material, resulting in a stereoselective production of (*S*)-(+)-MAMP [7].

Furthermore, only the enantiomer (*R*)-MDMA at <LOQ was detected in S5. This sampling point is located near effluent discharge of Crestuma WWTP showing that presence of this PAS may be correlated with effluent discharge. Of note, synthesis of MDMA (which is illicit in Europe) produces racemate MDMA. Nevertheless, after consumption (*S*)-MDMA undergoes preferential metabolism which leads to enrichment of (*R*)-MDMA (and subsequent excretion), as corroborated by the present results. Similar results were found in other studies [6,7,26]. At sampling points S2 and S3 AMPs were not detected (<LOD). These sampling points have been selected based on previous studies due to contamination by metals and other pollutants. Nevertheless, occurrence of PAS was not found at these sampling points corroborating that source of PAS may be related to WWTP effluent discharges.

Regarding SCAT, one enantiomer of BPD (D1) was found in both S3 and S4 in concentration <LOQ. Neither BPD D2 or 3-MMC, 3,4-DMMC and BTL were found at Douro River estuary.

Furthermore, 24 h-composite samples from two WWTPs (WWTPA and WWTPB), which effluents discharge into Douro River tributaries, were also analyzed. Regarding the concentration of AMPs in these effluents, (*R*)-AMP was found at <LOQ in WWTPA. The presence of (*R*)-AMP may indicate consumption of the racemate as human metabolism and the microbial processes during WWTP favors (*S*)-AMP.

(S)-MAMP was found in concentrations between <LOD and 57.30 ng $L^{-1}$ in WWTPA (Figure 9). This result corroborates estuarine water samples results that showed the presence of (*S*)-MAMPThis WWTP discharges into river Sousa at sampling point S4. Similar results were found in a previous report by Gonçalves et al. (2019) (25.7 ng $L^{-1}$) and in Albany (3.82–6.22 ng $L^{-1}$) [7,27]. Nevertheless, these values are lower than those reported in Vietnam (120–420 ng $L^{-1}$), U.S.A. (700 ng $L^{-1}$), Brazil (55.3–477.4 ng $L^{-1}$), and China (179 ng $L^{-1}$) [7,27–31].

Regarding to the MAMP EF, it is in accordance with that observed across the Europe (EF ≅ 1), with enrichment of (*S*)-(+)-MAMP. In fact, (*S*)-(+)-MAMP is considered a chiral signature of the European MAMP illegal market, besides being also the most reported in several other studies worldwide [6,7,26,32,33].

Regarding MDMA, concentrations were also found in range of <LOD to <LOQ, somewhat similar to those found across the Europe, i.e., <LOQ 3.2 ng $L^{-1}$ in Greece, 21.7 ng $L^{-1}$ in Portugal, <62 ng $L^{-1}$ in Croatia, and 0.5–24.8 ng $L^{-1}$ in U.K [7,34–36].

Although EF was not able to be determined in this study (concentrations <LOQ), it is known that MDMA metabolism is stereoselective favoring (*S*)-(+)-MDMA, with subsequent enrichment of (*R*)-(−)-MDMA excretion leading to enrichment of this enantiomer in the environment [7,34,37]. SCAT, BPD (D1) as well as 3,4-DMMC (D1) were found in range of <LOQ in WWTPA. Order of elution of enantiomers could not be determined as isolated enantiomers are not available. Nevertheless, results show that their presence occurs in different EF suggesting enantioselective processes. Separation of enantiomers is urgent for a comprehensive drug analysis.

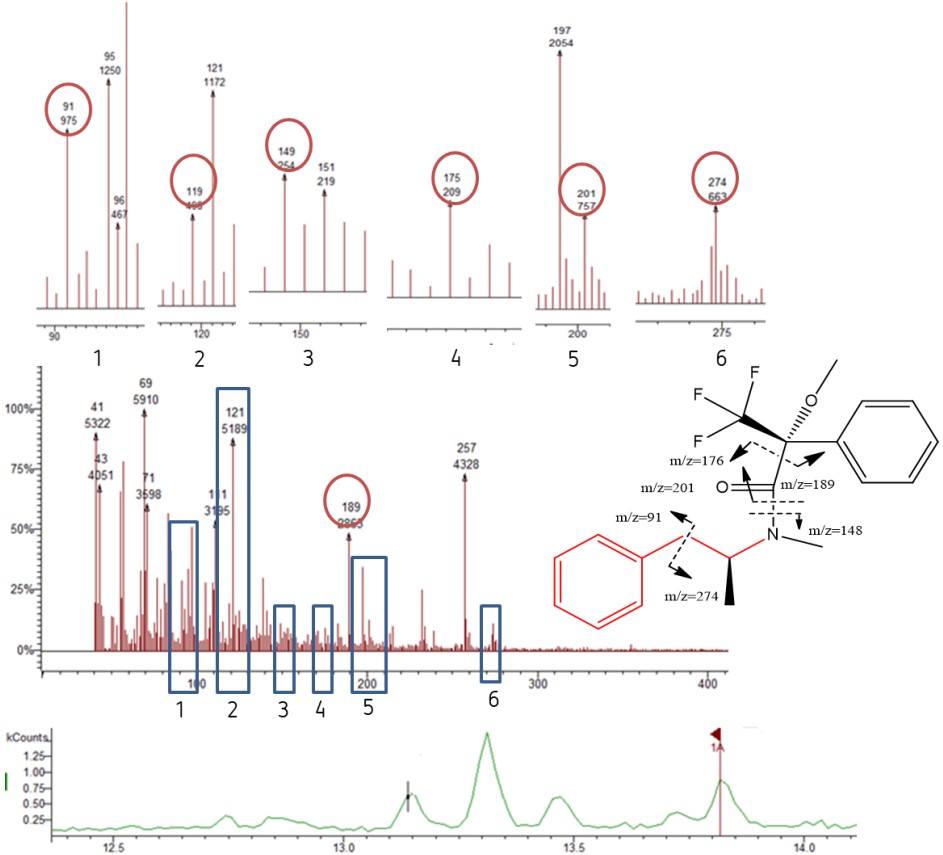

**Figure 9.** Chromatograms, mass spectrum and the suggested fragmentation pattern (*S*)-MAMP, as detected in WWTPA.

## 4. Conclusions

The main purpose of this study was to develop an enantioselective method by GC–MS for quantification of several classes of PAS in surface waters. The derivatization method with the enantiopure reagent (*R*)-MTPA-Cl was optimized to allow the formation of diastereomers of PAS including AMP (AMP, MAMP, and MDMA) and SCAT (BTL, 3,4-DMMC, 3-MMC, and BPD). Piperazines 1-BP and 1,4-MPP, although not chiral, were also derivatized with the chiral reagent, improving their signal identification and detection.

The chromatographic conditions were optimized to allow the validation of a method for the quantification of a total of 16 diastereomers and two derivatives of the target PAS in surface waters, in less than 24.0 min. The method was validated according to the ICH and showed to be linear ($r^2 > 0.98$), precise (0.40–7.83%), and accurate (82.4 to 116.9%). The LOD (14.2 to 89.5 ng L$^{-1}$) and LOQ (50 to 250 ng L$^{-1}$) allowed the method application for the detection and quantification of the target PAS in real environmental samples.

As such, the validated method was applied to investigate the occurrence and spatial distribution of the selected PAS in Portuguese surface waters in the Greater Porto region (AMP, (*R*)-MDMA, (*S*)-MAMP and the first enantiomer of BPD (configuration not assigned) and effluent samples from two WWTPs with different treatment technologies (BPD and 3,4-DMMC), along with other amphetamine derivatives.

The results obtained in this study allow to confirm that PAS are continuously consumed and discharged into the environment, being a potential threat for non-target organisms. The most common detected PAS were AMP, MAMP, MDMA, BPD, and 3,4-DMMC. Though sewage samples were not used in this study, enantioselective analysis of surface waters and effluent samples allowed to give insights about drug trends and consumption pattern in this specific region. Additionally, disclosure of their EF allowed discrimination between consumption and direct disposal and synthesis pathways. This study showed the environmental presence of these PAS and that they occur at different enantiomeric

mixtures. Therefore, enantioselective ecotoxicity studies should be done for an accurate risk assessment. Furthermore, isolation of enantiomers of SCAT is needed for a comprehensive analysis of PAS. This method showed its potential to monitor the selected PAS and will be applied for further studies to understand the pattern of consumption of these drugs in this region and for determination of the levels of these PAS in surface waters for environmental risk assessment studies.

**Supplementary Materials:** The following are available online at https://www.mdpi.com/article/10.3390/chemosensors9080224/s1; Table S1-Chemical structures, p*K*a and Log K_{ow} of the selected PAS. Table S2-Range of concentrations (ng L$^{-1}$) of the method calibration curve. Table S3-Quality control (QC) concentrations (ng L$^{-1}$) used for determination of accuracy, intra and inter-precision and recovery of the method. Table S4-Products of the reaction of the enantiopure derivatization reagent (R)-MTPA-Cl with AMP and AMP-type substances, SCAT, NK and PP. Table S5-Water temperature and physicochemical parameters (pH, EC and TDS) of Douro estuarine water samples. Table S6-Concentration and enantiomeric fraction (EF) for target PAS in effluent and Douro River estuarine water samples.

**Author Contributions:** Conceptualization, C.R. and M.E.T.; methodology, validation, I.L. and C.R.; formal analysis, I.L., D.S., C.R. and M.E.T.; writing—original draft preparation, I.L.; writing—review and editing, I.L., D.S., C.R. and M.E.T.; supervision, C.R. and M.E.T.; project administration, C.R. and M.E.T.; funding acquisition, C.R. and M.E.T. All authors have read and agreed to the published version of the manuscript.

**Funding:** This work was supported by National Funds through the Portuguese Science Foundation, FCT, I.P., project PTDC/CTA-AMB/6686/2020. Also supported by CESPU through the project MYCOBIOENV-PFT-IINFACTS-2019.

**Institutional Review Board Statement:** Not applicable.

**Informed Consent Statement:** Not applicable.

**Data Availability Statement:** All data generated and analyzed are included in the published article, supplementary material and upon reasonable request to the corresponding authors, C.R. and M.E.T.

**Acknowledgments:** The authors would like to thank Águas do Norte for the support and sample collection at the Water Treatment Plants.

**Conflicts of Interest:** The authors declare no conflict of interest.

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
