# Peer review of "Gas Chromatography Multiresidue Method for Enantiomeric Fraction Determination of Psychoactive Substances in Effluents and River Surface Waters"

_chemosensors, doi:10.3390/chemosensors9080224_

Round 1

Reviewer 1 Report

The paper “Gas chromatography multi-residue method for enantiomeric fraction determination of psychoactive substances in effluents and river surface waters” by I. Marcelino Langa et al. proposes an SPE-GC-MS method for some psychoactive substances in river waters and wastewater treatment plants effluents.

The paper is well designed, and good results are obtained. For those reasons, the manuscript could be suitable for publication after minor changes.

About the manuscript, I have some minor remarks to ask the authors.

Abstract: please, add the treatment of the sample on the abstract. Moreover, reduce the length of the two first paragraphs.

2.3. Sample collection.

Line 124. Explain why Leça river water was selected as a blank.

Line 138. Have you tested if the suspended particles contained any of the analyzed substances?

Line 142. Are the WWTPA samples obtained before or after the wastewater treatment? Have you tried to study the differences between both?

2.4. Solid-phase extraction (SPE) procedure.

Please, explain why you selected the OASIS MCX as the best extractive phase. In addition, because of the dual modes of retention (ion exchange and reversed-phase) of this phase, add in supplementary material a table with the log Kow and pKa values for the analytes.

Line 162-164. I do not understand why your last step here is to reconstitute the extract with 250 microL of MeOH, if immediately, in the next section, you take 200 microL and evaporate them. Could it be due to a stability problem? Otherwise, you lose 20% of your sample in this step.

2.5. Derivatization with chiral derivatization reagent.

Line 173. Please, explain how you recover quantitatively 1,5 microL of organic phase without taking any water with it.

2.7. Method parameters and validation

Line 205. The range of concentrations for the analytes in the calibration line is very small (sometimes, the highest concentration is slightly higher than twice the lower concentration). It is not usual, and you should explain the why. In addition, with so small a concentration range, it is not easy to obtain good correlation coefficients.

Line 215. This problem is transferred to the QCs standard solutions, where sometimes the high control sample concentration is less than twice the lowest control sample concentration (so, they are not a control sample at all).

3.2. Optimization of the chromatographic separation of the diastereomers and piperazine derivatives.

Line 267. Only one overlap is discussed here but, in Figure 3 and Table 1, most overlaps occur, but the retention times do not fit between them (for example, D2(R)MAMP and D1BDP or D23-MMC, D2-3,4-DMMC, D1(R)-MDMA and D1-NK). Please clarify this point.

3.4. Method validation.

Lines 350-352. I never saw before values for LOQ so rounded. Usually, the LOQs are estimated from the calibration data and give different values for the different compounds.

Table 2: Calibration lines figures of merit: add uncertainty estimations for the intercept and slope values. Add the number of points in each calibration line. In line 205 you say you use triplicate values for each calibration point. You must specify if you use these three values for the calibration line, or the mean value.

3.4. Method validation

Line 361. How do you explain the great difference between recoveries for 3-MMC and 3,4-DMMC? The chemical differences between both molecules are very small.

3.5. Application of the method.

Please, add a table with the analyte concentrations found in the different sampling points.

  1. Conclusions.

Line 450. 24 min is the required time for the chromatography determination, but not the time of the complete method.

Author Response

Reviewer #1:

The paper “Gas chromatography multi-residue method for enantiomeric fraction determination of psychoactive substances in effluents and river surface waters” by I. Marcelino Langa et al. proposes an SPE-GC-MS method for some psychoactive substances in river waters and wastewater treatment plants effluents.

The paper is well designed, and good results are obtained. For those reasons, the manuscript could be suitable for publication after minor changes.

About the manuscript, I have some minor remarks to ask the authors.

Response:  The authors thank the reviewer for the appreciation of the work and careful reading of the manuscript. The authors also thank the reviewer for the specific comments to improve our manuscript that were taken into account in this revised manuscript. Please see all changes and improvements in this revised version.

Abstract: please, add the treatment of the sample on the abstract. Moreover, reduce the length of the two first paragraphs.

Response: The two first paragraphs were reduced, and treatment of the sample added to the abstract.  Please see, page 1:

 Line 19 to 22:

“Determination of psychoactive substances (PAS) and/or their metabolites in surface waters is crucial for environmental risk assessment and disclosure of their enantiomeric fractions (EF) allows discrimination between consumption, direct disposal, and synthesis pathways.”

Line 31 – 32:

“For that, water samples were pre-concentrated by solid phase extraction using OASIS® MCX cartridges, derivatized and further analyzed by GC-MS.”

2.3. Sample collection.

Line 124. Explain why Leça river water was selected as a blank.

Response:  Matrix-matched calibration is commonly used to compensate for matrix effects and to allow an accurate quantification of the analytes. This is usually performed using blank matrices. Therefore, spring water samples from the source of Leça river, were selected as blank matrices, i.e., free of the target micropollutants while been a similar matrix of the analysed samples for an accurate quantification of the analytes.

To clarify, the following was added to the text, please see page 3, line 142-143:

“(to compensate matrix effects and allow an accurate quantification of the analytes)”.

Line 138. Have you tested if the suspended particles contained any of the analyzed substances?

Response: This is a pertinent comment. In fact, adsorption of other classes of psychoactive drugs have already been reported. Nevertheless, the objective of this work was to quantify these compounds in surface water and effluents samples. Therefore, suspended particles were not considered. Nevertheless, future studies are being planned to include analysis of these compounds in suspended particles for a comprehensive study.

Line 142. Are the WWTPA samples obtained before or after the wastewater treatment? Have you tried to study the differences between both?

Response: Samples from WWTPA were obtained after wastewater treatment (effluent samples). The propose of the analysis of these effluent samples was to have information about possible sources of these compounds in surface waters. Affluent samples (before treatment) have already been used in other works of our group for wastewater-based epidemiology studies (WBE) that were not the propose of this study.

2.4. Solid-phase extraction (SPE) procedure.

Please, explain why you selected the OASIS MCX as the best extractive phase. In addition, because of the dual modes of retention (ion exchange and reversed-phase) of this phase, add in supplementary material a table with the log Kow and pKa values for the analytes.

Response: OASIS MCX cartridges are mixed mode cation exchange sorbent and therefore indicated for the extraction of bases (amines). Nevertheless, on a previous study of our group, different sorbents were investigated, please see Coelho et al 2019. Oasis MCX showed to be the most suitable sorbent regarding recovery and elimination of interferences. In fact, these compounds are all bases (amines). Water samples are acidified and therefore these bases are ionized and therefore able to be retained by this sorbent.

Further information regarding log Kow and pKa for the analytes were added in the supplementary material (please see Table S1).

Coelho, M. M.; Ribeiro, A. R. L.; Sousa, J. C.; Ribeiro, C.; Fernandes, C.; Silva, A. M.; Tiritan, M. E., Dual enantioselective LC–MS/MS method to analyse chiral drugs in surface water: Monitoring in Douro River estuary. J Pharm  Biom Anal 2019, 170, 89-101

Line 162-164. I do not understand why your last step here is to reconstitute the extract with 250 microL of MeOH, if immediately, in the next section, you take 200 microL and evaporate them. Could it be due to a stability problem? Otherwise, you lose 20% of your sample in this step.

Response: The extract is reconstituted with 250 microL of MeOH to guarantee the recovery of all extract of the SPE procedure. Then, these samples must by derivatized for the formation of the derivatives before been analysed by GC-MS. Therefore, no hydroxyl groups should be available as they can react with the derivatizing reagent. Thus, MeOH must be evaporated. We reconstitute in 250 microL and take an aliquot of 200 microL to allow an accurate measurement of the sample volume and after derivatization it is reconstitute in the same volume (200 microL of anhydrous ethyl acetate and thus, the final concentration is the same.

2.5. Derivatization with chiral derivatization reagent.

Line 173. Please, explain how you recover quantitatively 1,5 microL of organic phase without taking any water with it.

Response: The authors apologize, there was an error in the text, is not 1.5 microL but 1500 microL. In fact, we use 1500 microL, and we do not recovery 1500 microL of the organic phase. To clarify the following was added to the text, please see page 5, line, 200:

“Then, 1200 µL of the organic phase was transferred…”

2.7. Method parameters and validation

Line 205. The range of concentrations for the analytes in the calibration line is very small (sometimes, the highest concentration is slightly higher than twice the lower concentration). It is not usual, and you should explain the why. In addition, with so small a concentration range, it is not easy to obtain good correlation coefficients.

Response: The range of concentration for calibration line is small. Nevertheless, higher ranges, or higher differences between concentrations are not justified because the expected environmental concentrations are very low. Therefore, low differences between selected concentrations for construction of the calibration curves of analytes for environmental samples are usually chosen, please see other works (Gonçalves et al 2019; Coelho et al 2019). In our study, good correlation coefficients were possible to obtain for all compounds.

Coelho, M. M.; Ribeiro, A. R. L.; Sousa, J. C.; Ribeiro, C.; Fernandes, C.; Silva, A. M.; Tiritan, M. E., Dual enantioselective LC–MS/MS method to analyse chiral drugs in surface water: Monitoring in Douro River estuary. J Pharm  Biom Anal 2019, 170, 89-101

Gonçalves, R.; Ribeiro, C.; Cravo, S.; Cunha, S. C.; Pereira, J. A.; Fernandes, J.; Afonso, C.; Tiritan, M. E., Multi-residue method for enantioseparation of psychoactive substances and beta blockers by gas chromatography–mass spectrometry. J Chromatogr. B 2019, 1125, 121731

Line 215. This problem is transferred to the QCs standard solutions, where sometimes the high control sample concentration is less than twice the lowest control sample concentration (so, they are not a control sample at all).

Response: The three quality controls should be within linear range (low, medium and high) for all analytes. Therefore, since range of calibration curve is small (due to the low levels found for these analytes in environmental samples), differences between concentrations of the calibration curve and QC are low. Nevertheless, an accurate method should be able to differentiate both calibration and QC concentration values.

3.2. Optimization of the chromatographic separation of the diastereomers and piperazine derivatives.

Line 267. Only one overlap is discussed here but, in Figure 3 and Table 1, most overlaps occur, but the retention times do not fit between them (for example, D2(R)MAMP and D1BDP or D23-MMC, D2-3,4-DMMC, D1(R)-MDMA and D1-NK). Please clarify this point.

Response: The authors apologize, in fact augmentation of the individual chromatograms may seem that there are more overlaps, but as the reviewer can see in Figure 8 (selectivity), only coelution of AMP/AMP d3 (IS) and BPD D2/3-MMC D1 occurs. Nevertheless, to not cause confusion Figure 3 was substitute for other, please see the new figure 3 on page 9.

3.4. Method validation.

Lines 350-352. I never saw before values for LOQ so rounded. Usually, the LOQs are estimated from the calibration data and give different values for the different compounds.

Response: We agree with reviewer comment. However, the first calibration level included in the calibration curve of all analytes was the lowest concentration able to be accurately quantified. The LOQ estimated by the calibration curve was the same or similar to the first level of the calibration curve, therefore, estimated LOQ values were used as the first levels for the construction of the CC and concentrations rounded for simplicity of the preparation of the working standards mixtures. For instance, an estimated LOQ of 49.7 ng/L was rounded to 50.0 ng/L.

Table 2: Calibration lines figures of merit: add uncertainty estimations for the intercept and slope values. Add the number of points in each calibration line. In line 205 you say you use triplicate values for each calibration point. You must specify if you use these three values for the calibration line, or the mean value.

Response: The uncertainty for the intercept and slope values were added to table 2. Regarding each calibration point these values can be found in supplementary material in Table S2.

3.4. Method validation

Line 361. How do you explain the great difference between recoveries for 3-MMC and 3,4-DMMC? The chemical differences between both molecules are very small.

Response: Differences between recoveries for 3-MMC and 3,4-DMMC can be due to interferences of the matrix. Also, though compounds are chemically similar, recoveries can be very different among substances, please see others works of our group:

Coelho, M. M.; Ribeiro, A. R. L.; Sousa, J. C.; Ribeiro, C.; Fernandes, C.; Silva, A. M.; Tiritan, M. E., Dual enantioselective LC–MS/MS method to analyse chiral drugs in surface water: Monitoring in Douro River estuary. J Pharm  Biom Anal 2019, 170, 89-101

Gonçalves, R.; Ribeiro, C.; Cravo, S.; Cunha, S. C.; Pereira, J. A.; Fernandes, J.; Afonso, C.; Tiritan, M. E., Multi-residue method for enantioseparation of psychoactive substances and beta blockers by gas chromatography–mass spectrometry. J Chromatogr. B 2019, 1125, 121731

3.5. Application of the method.

Please, add a table with the analyte concentrations found in the different sampling points.

Response: A table with the analyte concentrations in effluent and estuarine water samples was added in the supplementary material, please see Table S6.

Conclusions.

Line 450. 24 min is the required time for the chromatography determination, but not the time of the complete method.

Response: The authors agree, the sentence referrer to the chromatographic conditions, please see: “The chromatographic conditions were optimized to allow the validation of a method for the quantification of a total of 16 diastereomers and two derivatives of the target PAS in surface waters, in less than 24.0 min.”

Reviewer 2 Report

The authors quantitated psychoactive substances in river waters using a derivatization using (R)-(-)-α-methoxy-α-(trifluoromethyl) phenylacetyl chloride as chiral derivatization reagent. The article could be published after revision.

  1. It will be good to explain further, may be in the last paragraph of introduction, how did you select the specific 10 substances (amphetamine, methamphetamine, 3,4-methylenedioxymethamphetamine, norketamine, buphedrone , butylone, 3,4-dimethylmethcathinone, 3-methylmethcathinone, 1-benzylpiperazine and 1-(4-metoxyphenyl)-piperazine) for quantitation? Are there any other methods reported in literature for the enantioselective determination of these compounds?
  2. Please explain in detail the novelty of the proposed method over the already described methods reported here: 1. derivatization procedure described by Gonçalves et al (2019), and 2. SPE procedure was adapted from that described by Coelho et al [15].
  3. Please explain all the acronyms used in section 2.5 and 2.7 , (e.g. 0.02% TEA in Hex, etc)
  4. Table 1S, and Table 2S, Table2, how did you choose the selected concentration range for method validation?
  5. Line 226, It is not quite clear to me, why did you use only peak area for NK and PP?
  6. Line 228 : please explain EF ….
  7. Figure 2 should be presented in one page and in a better resolution
  8. Application to real samples: the text needs to be more comprehensive and it should be seriously reduced.
  9. Overall the text and organization of the manuscript should be improved, in method development-method validation and application.

Author Response

Reviewer 2

The authors quantitated psychoactive substances in river waters using a derivatization using (R)-(-)-α-methoxy-α-(trifluoromethyl) phenylacetyl chloride as chiral derivatization reagent. The article could be published after revision.

Response: The authors thank the careful reading and valuable suggestions that were taken into account in this revised manuscript. Please see all changes and improvements in this revised version

It will be good to explain further, may be in the last paragraph of introduction, how did you select the specific 10 substances (amphetamine, methamphetamine, 3,4-methylenedioxymethamphetamine, norketamine, buphedrone, butylone, 3,4-dimethylmethcathinone, 3-methylmethcathinone, 1-benzylpiperazine and 1-(4-metoxyphenyl)-piperazine) for quantitation? Are there any other methods reported in literature for the enantioselective determination of these compounds?

Response: The authors thank the reviewer comment, and the following was added to the introduction, please see page 2, line 82-84:

“These substances were selected based on EMCDDA reports and literature regarding the consumption of NPS [1-2, 18].

To the best of our knowledge, there are not reports for the enantioselective determination of all these classes of compounds in surface waters.”

Please explain in detail the novelty of the proposed method over the already described methods reported here: 1. derivatization procedure described by Gonçalves et al (2019), and 2. SPE procedure was adapted from that described by Coelho et al [15].

Response: The previous methods reported by Gonçalves et al and Coelho et al did not included the synthetic cathinones and piperazines. However, as synthetic cathinones and piperazine are amines, the same protocols of sample preparation and derivatization were adapted to include these new compounds.

Please explain all the acronyms used in section 2.5 and 2.7 , (e.g. 0.02% TEA in Hex, etc)

Response: The authors think that a misunderstood must have occurred, all these acronyms are explained in previous sections, please see section 2.1 (Chemicals and materials).

Table 1S, and Table 2S, Table2, how did you choose the selected concentration range for method validation?

Response: The target analytes are present at residual concentrations in environmental matrices; therefore, selected range of concentrations were based on the lowest quantifiable with accuracy and precision (LOQ) and higher concentrations to be possible to construct a calibration curve for each substance.

Line 226, It is not quite clear to me, why did you use only peak area for NK and PP?

Response: The authors clarify, for those compounds no internal standard, therefore only peak areas were used for the calibration curves.

Line 228 : please explain EF ….

Response: EF refers to the enantiomeric fraction (EF). The acronymous is previous explain in the introduction. Please see page 2, line 62.

Figure 2 should be presented in one page and in a better resolution

Response: A better resolution of Figure 2 in added to the manuscript, nevertheless, due to chemosensors template is not possible to present the figure in only one page.

Application to real samples: the text needs to be more comprehensive and it should be seriously reduced.

Overall the text and organization of the manuscript should be improved, in method development-method validation and application.

Response: The application to real samples was revised and reduced please see page 18.

Reviewer 3 Report

The introduction provides enough information to understand the importance of detect psychoactive substances and their derivatives and how their detection on residual waters can be used in order to estimate the population consumption with important consequences. This section includes the analytical techniques commonly used for the analysis of psychoactive substances and clarifies the objective of this research.

The material and methods section is correctly divided and it provides enough information to reproduce the current research. A point that I consider of high importance for this kind of article. Just as a comment, in line 212 the LOQ and LOD are in different formats. Please, unify them.

I consider that the results obtained from this article are of high quality, the authors have achieved the development of a method for the detection and quantification of several classes of psychoactive substances in surface waters. The developed method has exhibited a high selectivity and accuracy and it has been applied in real samples with really good results. In addition, the conclusions are supported by the results obtained.

The English I consider of high quality but I highly recommend do not use the word "work" in this kind of research because I consider that it is not totally correct. The authors used it several times during the article (lines 231, 419, 430, 459....). I highly recommend using another expression as research, investigation, article, project, etc.

For last, the bibliography employed is actual and is in the correct format. Although, I recommend including the DOI when it was possible.

I consider this article of high quality and importance for the scientific community and I recommend its publication.

Author Response

The introduction provides enough information to understand the importance of detect psychoactive substances and their derivatives and how their detection on residual waters can be used in order to estimate the population consumption with important consequences. This section includes the analytical techniques commonly used for the analysis of psychoactive substances and clarifies the objective of this research.

The material and methods section is correctly divided and it provides enough information to reproduce the current research. A point that I consider of high importance for this kind of article. Just as a comment, in line 212 the LOQ and LOD are in different formats. Please, unify them.

I consider that the results obtained from this article are of high quality, the authors have achieved the development of a method for the detection and quantification of several classes of psychoactive substances in surface waters. The developed method has exhibited a high selectivity and accuracy and it has been applied in real samples with really good results. In addition, the conclusions are supported by the results obtained.

The English I consider of high quality but I highly recommend do not use the word "work" in this kind of research because I consider that it is not totally correct. The authors used it several times during the article (lines 231, 419, 430, 459....). I highly recommend using another expression as research, investigation, article, project, etc.

For last, the bibliography employed is actual and is in the correct format. Although, I recommend including the DOI when it was possible.

I consider this article of high quality and importance for the scientific community and I recommend its publication.

Response: The authors greatly appreciated the careful reading and all valuable comments of our study. Please also see all changes and improvements in this revised version to the specific comments.

Specific comments:

  1. A point that I consider of high importance for this kind of article. Just as a comment, in line 212 the LOQ and LOD are in different formats. Please, unify them

Response: The authors thank the reviewer and formats were uniformized.

  1. The English I consider of high quality but I highly recommend do not use the word "work"

Response: The word “work” was replaced by “study” or “report” along the manuscript.

  1. For last, the bibliography employed is actual and is in the correct format. Although, I recommend including the DOI when it was possible.

Response: The DOI was added to the references when available, please see references.